# Automated Hypertension Detection Using ConvMixer and Spectrogram Techniques with Ballistocardiograph Signals

**DOI:** 10.3390/diagnostics13020182

**Published:** 2023-01-04

**Authors:** Salih T. A. Ozcelik, Hakan Uyanık, Erkan Deniz, Abdulkadir Sengur

**Affiliations:** 1Electrical-Electronics Engineering Department, Engineering Faculty, Bingol University, Bingol 12000, Turkey; 2Electrical-Electronics Engineering Department, Engineering Faculty, Munzur University, Tunceli 62000, Turkey; 3Electrical-Electronics Engineering Department, Technology Faculty, Firat University, Elazig 23119, Turkey

**Keywords:** hypertension, high blood pressure, BCG signal, spectrogram, convolutional mixer

## Abstract

Blood pressure is the pressure exerted by the blood in the veins against the walls of the veins. If this value is above normal levels, it is known as high blood pressure (HBP) or hypertension (HPT). This health problem which often referred to as the “silent killer” reduces the quality of life and causes severe damage to many body parts in various ways. Besides, its mortality rate is very high. Hence, rapid and effective diagnosis of this health problem is crucial. In this study, an automatic diagnosis of HPT has been proposed using ballistocardiography (BCG) signals. The BCG signals were transformed to the time-frequency domain using the spectrogram method. While creating the spectrogram images, parameters such as window type, window length, overlapping rate, and fast Fourier transform size were adjusted. Then, these images were classified using ConvMixer architecture, similar to vision transformers (ViT) and multi-layer perceptron (MLP)-mixer structures, which have attracted a lot of attention. Its performance was compared with classical architectures such as ResNet18 and ResNet50. The results obtained showed that the ConvMixer structure gave very successful results and a very short operation time. Our proposed model has obtained an accuracy of 98.14%, 98.79%, and 97.69% for the ResNet18, ResNet50, and ConvMixer architectures, respectively. In addition, it has been observed that the processing time of the ConvMixer architecture is relatively short compared to these two architectures.

## 1. Introduction

The word “hypertension” describes a common medical condition in which blood pressure on the walls of arteria as it flows from the heart to the body is too high to prevent health problems, including heart disease. It can go unnoticed for many years by many people. According to the World Health Organization, there are now 1.28 billion persons worldwide who have hypertension [1]. It can severely harm the kidney, brain, heart, and vascular system and go unnoticed for a long period [1]. The early diagnosis of hypertension is critical because of the great damage it can cause to the human body. The determination of the amount of blood passing through the heart by measuring the reactions in the body as a result of the sudden ejection of blood from the heart as a result of each cardiac cycle is called ballistocardiography (BCG) [2]. Many methods have been developed to detect and treat hypertension by utilizing various physiological signals other than BCG. Prominent physiological signals used are electrocardiogram (ECG), seismocardiography (SCG), forcecardiography (FCG), and gyrocardiography (GCG). Parmar et al. [3] used the Fourier decomposition method (FDM) and a uniform cosine modulated filter bank (CMFB) to divide the ECG signal into eight Fourier intrinsic band functions (FIBF’s). After decomposition, they extracted the signal mobility and log-energy entropy properties for the decomposed sub-signals. They obtained the highest detection accuracy of 99.11% with the k-NN classifier. Soh et al. [4] proposed a model to detect hypertension automatically using ECG signals. They used the normal sinus rhythm dataset recorded at Massachusetts Institute of Technology-Beth Israel Hospital (MIT-BIH) [5] and the Smart Health for Assessing the Risk of Events via ECG (SHAREE) [6] dataset. The ECG signals were denoised using discrete wavelet transform (DWT), and then divided into frequency bands using the high pass and low pass filters. They reported a detection accuracy of 99.99% using a convolutional neural network (CNN) with ten-fold cross-validation. Using the SHAARE dataset, Rajput et al. [7] extracted fractal dimension and log entropy features from the ECG signals. They developed a new hypertension diagnostic index to discriminate between hypertension and low blood pressure patients. Using SHAARE dataset, Jain et al. [8] developed a two-stage deep learning-based classification model. Their system could distinguish between hypertension and normal patients with a success rate of 99.68%. BCG determines the effect of this muscular activity by directly measuring the acceleration of blood. While the BCG records the mechanical vibrations based on heart action, an ECG measures cardiac electrical activity [9]. The BCG is preferable to ECG because it does not involve the installation of electrodes into the patient’s body, which interferes with the subject’s usual sleep patterns during data collection [10]. The analysis of heart rate variability, the exploration of cardiac anomalies, and the diagnosis of hypertension have become crucial techniques in biomedical research because conventional methods of detecting hypertension cannot diagnose a few disorders. Song et al. [11] continuously obtained BCG signals with micro motion sensitive mattress (MSM) during sleep in the dataset they created with eight women and 10 men. They proposed a method based on ensemble empirical mode decomposition (EEMD) in their work. They extracted the heart rate variability (HRV) features and time domain, frequency domain, and nonlinear features from the BCG signals. They achieved a 92.3% classification accuracy using a Naive Bayes classifier. Liu et al. [12] recorded BCG signals during the night’s sleep in an experiment created using 61 hypertension and 67 normotensive people. They extracted sample entropy, approximate entropy, HRV time and frequency domain, and BCG fluctuation features from BCG signals. Their studies obtained 84.4% accuracy, 82.5% precision, and 85.3% recall in detecting hypertension patients. Rajput et al. [13] converted BCG signals into scalogram images using continuous wavelet transform (CWT) to detect individuals with hypertension automatically. They used a 2D convolutional neural network model and achieved an accuracy rate of 86.14% using dataset [12]. In a recent study, Gupta et al. [14] developed a method to automatically detect patients with hypertension using the dataset created in [12]. They split the BCG signals into sub-bands using the integrated tunable Q factor wavelet transform (TQWT) combined with a multi-verse optimization (MVO) algorithm. They extracted 11 statistical features: Shannon entropy, log energy, Hjorth complexity, standard derivation, root mean square value, kurtosis, skewness, mean value, maximum value, and minimum value in each band. They achieved a 92.21% classification accuracy rate using the kNN classifier with a 10-fold cross-validation strategy. Gupta et al., in another recent study [15], developed an automatic hypertension detection system by converting BCG signals to time-frequency (T-F) spectral images. Again, they used the dataset produced in [12]. Gabor transforms (GT), smoothed pseudo-Wigner Ville distribution (SPWVD), and short Fourier transform (STFT), were used to obtain T-F spectral images. They obtained an accuracy of 97.65% with 10-fold cross-validation with their proposed CNN model (Hyp-Net). Seok et al. [16] prepared a dataset that measured chair-based BCG signals with 30 participants. In the first part of the experiment, which consisted of two parts, two-channel BCG signals and blood pressure values were recorded. The second part repeated the same measurements after the subject ran on the treadmill. To eliminate noise from input signals and establish an instantaneous phase for model training, they used the Hilbert transform with the empirical mode decomposition (EMD) method. The authors compared the results of the first session (rest) and the second session (recovery) after training a CNN regression model that predicted systolic and diastolic blood pressures (SBP and DBP) from the two-channel BCG phases. In another study, Rajput et al. [17] used BCG signals to diagnose normotensive and hypertensive subjects. Using the dataset in [12], they split the BCG signals into 30 s segments using EMD and wavelet transform (WT). Using WT and EMD, the BCG signal was divided into five sub-bands and five levels of intrinsic mode functions (IMFs). They used the ensemble gentle boost classifier to classify the retrieved features. The researchers achieved an 89% classification accuracy rate using 10-fold cross-validation for WT.

Our present study proposed a novel approach for BCG-based hypertension detection. The input BCG signals were normalized using the median absolute derivation approach and then filtered using a band pass filter frequency range between 1 and 50 hertz. The Chebyshev type II filter was used to filter the BCG signals. Then, a non-overlapping signal segmentation procedure was applied to divide the input BCG signals into multiple BCG signals. The length of each segment was 30 s, and each signal was then converted to a heat map image using the spectrogram approach. After obtaining the BCG heat map images, the feature extraction and classification operations were performed using ConvMixer architecture. ConvMixer is a simplified CNN approach where the input images are divided into non-overlapping square patches. The obtained patches are embedded to form a tensor that will go through the ConvMixer architecture. Various depths of the ConvMixer architecture were designed and applied to the BCG signal dataset. The BCG dataset is comprised of 128 subjects. In all, 61 of the 128 subjects are labeled as hypertensive (HPT), and the rest, 67 subjects, are labeled as normotensive (NRT). We have used various statistical measures to evaluate the model’s performance. The obtained results were also compared with the existing approaches. The main contributions of this work are as follows:This is the first time ConvMixer architecture has been applied to BCG-based hypertension detection.The developed model is simple and computationally less complex.

The paper organization is as follows: In Section 2, the used dataset explained. In Section 3, the proposed methodology scheme and its application steps are provided. In Section 4, the experimental works and results are interpreted in detail. In Section 5, the advantages and disadvantages of the proposed method are emphasized and compared with other similar studies. Finally, the paper is concluded in Section 6 with conclusions.

## 2. Database

In this work, we used a publicly available dataset developed using a signal acquisition system called the RS-611 [12]. It is accessible from this link: https://doi.org/10.6084/m9.figshare.7594433 (accessed on 22 May 2022). In the signal acquisition system, the signals originating from the cardiac movements of the heart were recorded with a 16-bit analog-to-digital converter system at a 100 Hz sampling rate. The dataset is composed of 61 subjects with hypertension (HPT) and 67 subjects with normal blood pressure (NRT). The details of the dataset are shown in Table 1.

## 3. Methodology

A flowchart of the methodology used in this study is illustrated in Figure 1. A series of preprocessing steps are applied to the BCG signals to make them convenient for the subsequent stages of the proposed work. Hence, a normalization procedure is performed where the amplitude of the raw BCG signals is centered and scaled with z-score normalization to have a 0 median and median absolute deviation to be 1. The normalization of raw BCG signals is mandatory because the differences in the subjects’ body weights directly affect the BCG signals [18]. Besides, a fourth-order bandpass Chebyshev type II filter is designed to denoise the normalized BCG signals. The frequency range of the bandpass filter is 1 to 50 Hz. The preprocessed BCG signals are then segmented by using a non-overlapping fixed-length size sliding window, which is frequently applied in the signal processing field. The length of the signal segmentation window is 30 s.

In the next step, spectrogram images of the filtered BCG signals are obtained. This process aims to obtain the most appropriate spectrogram image with some parameter preferences (window type, window length, overlap ratio, number of FFTs to be applied, and threshold value selection). As previous studies [19,20,21] suggested, the balance of time-frequency resolution is essential to obtain the spectrogram images.

In the last step, the spectrogram images of BCG signals are classified using the ConvMix classifier. Then the results obtained were compared with ResNet18 and ResNet50 classifiers.

### 3.1. Normalization

All the BCG signals are normalized to prevent the differences in signal values that may occur due to differences in body weights. In this work, we have used Z-score normalization. The median absolute deviation (MAD) was employed while applying Z-score normalization. The MAD method is preferred instead of the standard deviation when there are high differences in the value distribution in the data to be normalized. The high differences in the value distribution within the data distort the standard deviation calculation. This is not the case in the MAD method [22].

The MAD of a dataset is the median of the absolute deviations of the data from the median of the data ($\tilde{X\;}$). It is calculated as in the following equation [22].
(1)$$\mathrm{MAD}=\mathrm{median}\left(\;\left|\;x-\tilde{X\;}\right|\right)\;$$

Figure 2 shows the normalization process of an example signal in the BCG signal dataset.

### 3.2. Filtering

The BCG signal used in this study was filtered with a passband of 1–50 Hz fourth order of Chebyshev type II IIR filter. The filter was created with the filter designer toolbox in MATLAB. Figure 3 shows the magnitude response of the used Chebyshev type II filter.

The coefficients of the designed filter are as follows:

Numerator Coefficients = (0.028 0.053 0.071 0.053 0.028)

Denominator Coefficients = (1 −2.026 2.148 −1.159 0.279)

### 3.3. Spectrogram

The processing of signals in the time-frequency domain is one of the most common methods in the signal-processing field. The most important details about when and how the spectral information of the signals processed in the time-frequency domain changes. Such information cannot be observed in linear time-invariant (LTI) operations such as the Fourier transform. The spectrogram method provides the best way to observe the spectral information of the signal in the time-frequency domain. The spectrogram is frequently used in the spectral analysis of different types of non-stationary signals, such as biomedical [23,24,25], speech [26], music [27], and seismic signals [28], which calculates the Fourier transform of the signal with the help of a sliding window.

In this study, the Hanning window was used because of its many advantages in the signals’ windowing step. The most important advantage of this window is it has a narrow main lobe and similarly narrow and fast dumping side lobes in the frequency domain. This feature prevents the undesired situation called spectral leakage, which causes the weakening of spectral information. The window length was chosen with a sample length of 128. The overlapping ratio, an important parameter in observing repeating features, was chosen as 15/16. Thus, data losses that may occur in short intervals in the BCG signals are reduced. A disadvantage of choosing this parameter high in this way is that it increases the processing load required when acquiring the spectrogram images. Another preference made while obtaining spectrogram images of BCG signals is to observe better the amplitude value, which is the third dimension. In these images, a threshold value was applied to the images, and the points with low amplitude values were eliminated. The points with high amplitude values are more reliable in terms of spectral information than points with low amplitude values. In addition, a simplified image will give more successful results for the neural network classifiers to be used in the classification phase. Lastly, the FFT length of 128 was chosen. With these parameter preferences, the spectrogram image of a 3000-sample long signal belonging to a BCG signal in the dataset was formed as in Figure 4 and the 3D image in Figure 5.

As mentioned above, the results of the filtering process applied while obtaining the spectrogram images are shown in Figure 6 with an example.

### 3.4. Convolution Mixer (ConvMixer)

ConvMixer is an architecture similar to vision transformers (ViT) [29] and MLP-Mixer [30] structures. It directly uses patches as input, separating the mix of spatial and channel sizes and working with equal size and resolution across the network. However, unlike these two architectures, ConvMixer only performs standard convolution operations to perform mixing operations [31]. As seen in Figure 7, ConvMixer architecture comprises three building parts. In the first part is a patch embedding process where the input image is split into patches, and the obtained patches are embedded to form the tokens. Patch embedding converts an N × N image into a feature map of size h × n/p × n/p, where h symbolizes the hidden dimension and p × p is the size of the patch. After patch embedding, a Gaussian error linear unit (GeLu) layer and a batch normalization layer are used. The primary ConvMixer layer is repeated for a predetermined number of depth times in the second part of the model. A depth-wise convolution is contained in the residual block that makes up this layer. One layer’s output is combined with another layer’s output in a residual block of the ConvMixer layer. In this instance, the inputs are combined with the depth-wise convolution layer’s output. The activation block, a pointwise convolution, and another activation block are all placed after this output. A global pooling layer makes up the third part of the model, which produces a feature vector of size h that may be sent to a Softmax classifier or any other head, depending on the application.

## 4. Experimental Works and Results

All experimental works were conducted on MATLAB using signal processing and deep learning toolboxes. Two different computers were used in the experiment. The first computer has an Intel Core i5-12400F CPU, 16 GB RAM, and Nvidia RTX 2080 Ti GPU. The second one has dual Intel Xeon Gold 5120 CPUs, 32 GB RAM, and dual 4 GB Nvidia Quadro RTX 4000 GPUs. The BCG signals were filtered using a band pass filter, where the filter coefficients were adjusted using a band pass filter, and the filter coefficients were adjusted using MATLAB filter designer. The parameters for the spectrogram images were determined heuristically during initial experimental works. The spectrogram images had an initial size of 420 × 560 × 3 and, to make them compatible with ConvMixer, ResNet18, and ResNet50 architectures, they were resized to 224 × 224 × 3, respectively. For the ConvMixer architecture, the patch size = 5 × 5, hidden dimension = 32, and depth of the mixer = 7 were chosen. The ConvMixer was trained with the ‘sgdm’ optimizer where the minibatch size was set to 64, the number of the maximum epoch was set to 7, and the initial learning rate was chosen as 0.001. The ConvMixer architecture used in this study is shown in Figure 8, and the first convolution layer weights are shown in Figure 9. As seen in Figure 8, the depth of the architecture was 7, and residual connections were used to connect the batch normalization layers to the pointwise convolution layers in each ConvMixer block.

The ConvMixer convolution layer parameters were set as filter number 32, filter size [5 5], stride [5 5], and dilation factor [1 1], and the ConvMixer first convolution layer weights obtained are shown in Figure 9 after the classification process.

In this work, we considered two different strategies. In the first one, an equal number of spectrogram images from 128 subjects were considered. To handle it, the first 50 min of the BCG signals, which is the shortest BCG signal recording length among all dataset, were used for each subject. Thus, an equal number of spectrogram images (100 samples at 30 s signal segments) from each subject was used to obtain 12,800 spectrogram image sets (6100 HPT spectrogram images and 6700 NRT spectrogram images).

In the second experiment, we considered the entire dataset. Similar to the first step, BCG signals were divided into 30 s segments in this step. As a result, a total of 132,938 spectrogram images (HPT: 61,525 and NRT: 71,413) were obtained. These images were fed to ConvMixer, ResNet18, and ResNet50 networks for classification.

We randomly split the dataset into 10 parts. In the training process, 9 out of 10 parts were used for training, and the remaining one was used for testing. This procedure was implemented repeatedly until they were used once as testing parts. Using the ResNet architectures, we used transfer learning of these architectures. Then, input and output layers were adjusted to be compatible with the used spectrogram images and classes. The ‘sgdm’ optimizer was used to adjust the model’s parameters in all experiments.

Figure 10 shows the training progress of the ConvMixer architecture for a trial of the first experiment, which used 12,800 spectrogram images. As seen in Figure 10, the training and validation accuracies were around 90% at the end of the first epoch. And the accuracies were increased until the end of the 4th epoch. After 4th epoch, a bit of overfitting was observed, and the final validation accuracy was 93.44%.

Figure 11 and Figure 12 show that training and validation accuracy of 90% was obtained using ResNet architectures at the end of their first epoch. After 4th epoch, a bit of overfitting was observed during their training process. As a result, their final validation accuracies of 93.75% and 95.78% were obtained for ResNet18 and ResNet50, respectively. It may be noted that the amount of fluctuation in the training curve (blue line) of ConvMixer architecture is much less than in the other two architectures.

The performance evaluation metrics obtained for 12,800 spectrogram images with a ten-fold cross-validation strategy is given in Table 2. It may be noted from the table that the highest accuracy of 95.89% was achieved by fine-tuning the pre-trained ResNet50 model. The second-best average accuracy of 93.96% was achieved using the fine-tuned ResNet18 model. Conversely, the ConvMixer produced the least average accuracy score of 93.84% with 12,800 spectrogram images.

Figure 13 and Figure 14 show the confusion charts obtained for one-fold of the 10-fold cross-validation strategy and cumulative of the 10-fold cross-validation strategy, respectively. As seen in Figure 13 and Figure 14, both ResNet models produced better predictions than the ConvMixer model with 12,800 and 132,968 spectrogram image experiments.

Figure 15 shows the training progress of the ConvMixer architecture obtained for the second experiment with 132,968 spectrogram images. As seen in Figure 15, the training, and validation accuracies were over 92% at the beginning of the first epoch. The accuracies were increased until the end of the training process. Again, there was no observation about the overfitting situation, and the final validation accuracy of 96.89% was obtained.

Similarly, Figure 16 and Figure 17 show that the training and validation accuracies reached over 92% at the beginning of their first epoch for ResNet architectures. A bit of overfitting was observed during their training process. Similar to ConvMixer’s result, there was no overfitting situation, and the final validation accuracy of 98.10% and 98.39% was obtained for ResNet18 and ResNet50, respectively. The first experiment results indicate that the amount of fluctuation in the train’s curve (blue line) of ConvMixer architecture is again much less than the other two architectures.

The performance evaluation metrics obtained with ten-fold cross-validation for 132,968 spectrogram images are shown in Table 3. As seen in Table 3, both ResNet models produced better average evaluation scores than the ConvMixer model. Besides, the ResNet50 model produced better average evaluation scores than the ResNet18 model. The average accuracy scores obtained were 98.79%, 98.14%, and 97.69% for the ResNet50, ResNet18, and ConvMixer models, respectively.

Figure 18 and Figure 19 show the confusion charts obtained for three architectures with 132,968 spectrogram images during one-fold of the ten-fold cross-validation strategy and cumulative of the ten-fold cross-validation strategy, respectively. As seen in Figure 18 and Figure 19, both ResNet models produced better predictions than the ConvMixer model with 132,968 spectrogram images.

Our results show that the model tends to perform better with more images (second experiment).

## 5. Discussions

In this study, we introduced a novel scheme for classifying HPT versus healthy cases based on the BCG signals. After a series of preprocessing steps (discussed in the Methodology section), the BCG signals were transformed into time-frequency images based on the tuned spectrogram technique. Then a novel CNN model, namely ConvMixer, was adopted in the classification stage of the proposed study. We have used 12,800 and 132,968 spectrogram images in our work to develop the model. Various performance evaluation metrics were used to report the proposed model and fine-tuned two pre-trained CNN models with a 10-fold cross-validation strategy. The obtained results showed that the proposed ConvMixer architecture is quite performant on the used BCG signal dataset in this study. We have obtained an average accuracyof 93.84% and 97.69% for the proposed ConvMixer architecture, using 12,800 and 132,968 spectrogram images, respectively.

The classification results obtained using ten-fold cross-validation are shown in Table 2 and Table 3. It can be noted from these tables that our proposed model achieved accuracies close to the pre-trained CNN models for both numbers of spectrograms (12,800 and 132,968). This is because the ConvMixer architecture has a lower computational complexity than the traditional ResNet architectures. Most importantly, we have reached the highest accuracy with all methods (Table 2 and Table 3) with the BCG signal dataset. This proves that the spectrogram images can successfully extract the features to obtain the highest classification performances in the automated detection of HPT using BCG signals. In Table 4, we listed previous works on detecting HPT using similar BCG dataset. It may be noted that most works have employed machine learning techniques for HPT detection. Until now, the best accuracy score of 97.65% was obtained by Gupta et al. [15] using the CNN model. Authors in [11] obtained a 92.3% accuracy score using the CNN model, and Gupta et al. [14] obtained 92.21% with the same BCG database.

Authors in [13,17] used various wavelet transform-based features with traditional classifiers and obtained 86.14% and 89% accuracy scores, respectively. We have achieved the highest accuracy of 97.69%.

The advantages of our work are as follows:1-A simple, accurate, and efficient model is developed for hypertension detection using BCG signals.2-Most discriminative frequency components are extracted from the time-frequency image-based signal classification purposes domain using the spectrogram approach.3-This is the first study to use the ConvMixer model with a time-frequency image-based classification application for BCG signals.4-The ConvMixer model is computationally efficient when it takes a shorter training time than the traditional deep CNN models.

The disadvantages of this work are the parameter adjustment of ConvMixer and the spectrogram to obtain the optimum performance are arduous.

## 6. Conclusions

This study has developed an accurate and computationally less complex approach for automated hypertension detection using BCG signals. The input BCG signals were initially segmented and filtered in the proposed approach. Then, a fine-tuned spectrogram approach was used to convert the signals into heat map images. The ConvMixer was used for classification purpose. The proposed approach not only produced high accuracy scores in a short time as compared to other pretrained CNN models. The main limitation of this study is that we have used only 128 (61 HPT and 67 normal) subjects. We plan to use more subjects and validate our model in the future.

Additionally, we intend to use metaheuristic approaches to determine the optimum parameters for both the spectrogram approach and the ConvMixer model. Further, the proposed model can be used in various healthcare applications using physiological signals like an electroencephalogram (EEG), electromyogram (EMG) etc.

## Figures and Tables

**Figure 1 diagnostics-13-00182-f001:**
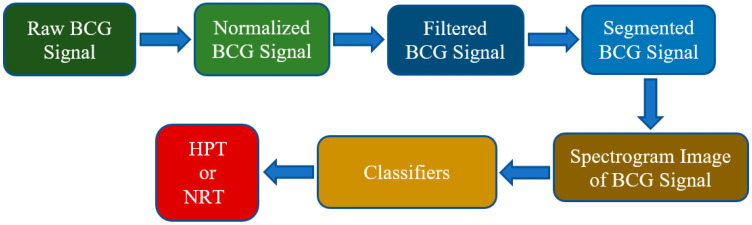
Proposed methodology for automatic detection of hypertension cases.

**Figure 2 diagnostics-13-00182-f002:**
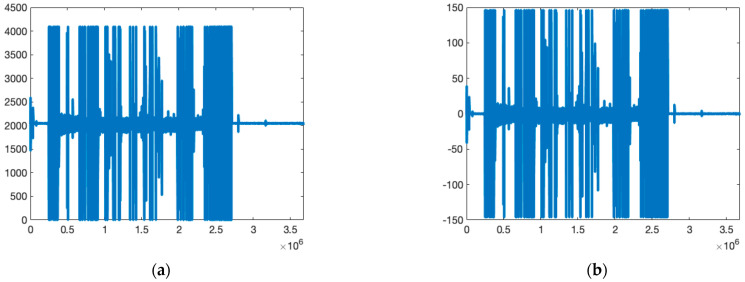
Normalization of BCG signals. (**a**) An example of raw BCG signal from the dataset. (**b**) Normalized BCG signal.

**Figure 3 diagnostics-13-00182-f003:**
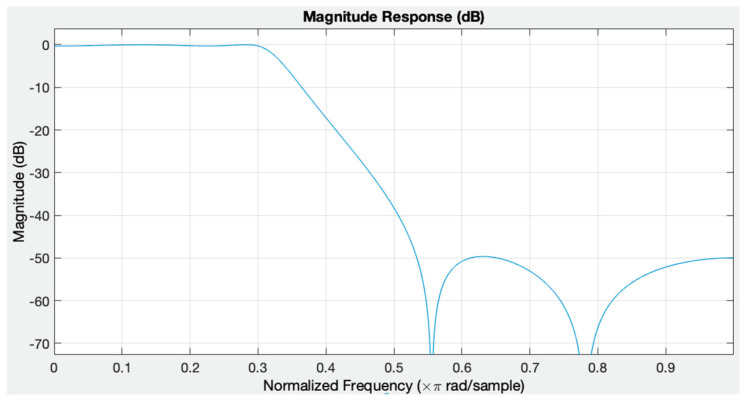
Magnitude response of the used filter.

**Figure 4 diagnostics-13-00182-f004:**
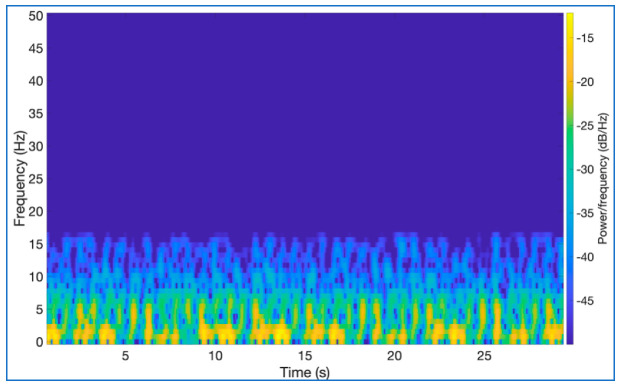
Spectrogram image of a BCG segment (30 s).

**Figure 5 diagnostics-13-00182-f005:**
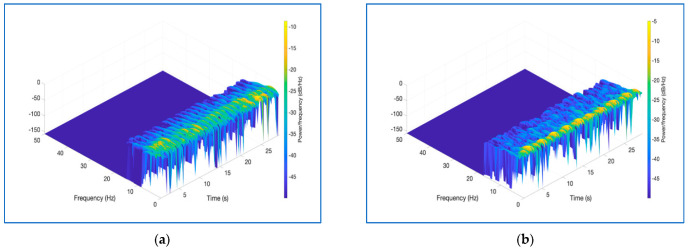
3D spectrogram image of normotensive (**a**) and hypertensive (**b**) subjects in the dataset.

**Figure 6 diagnostics-13-00182-f006:**
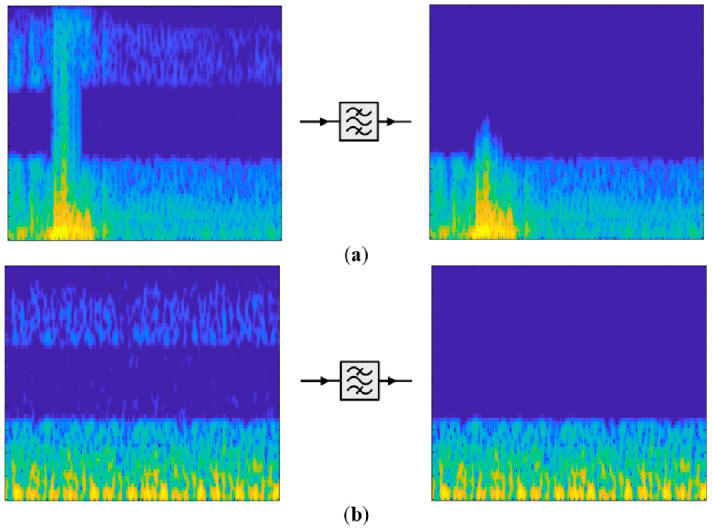
Examples of filtering effect: (**a**) HPT case example. (**b**) An NRT case example.

**Figure 7 diagnostics-13-00182-f007:**
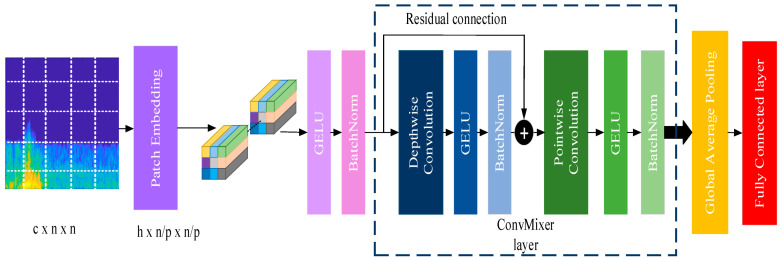
Convolution mixer structure.

**Figure 8 diagnostics-13-00182-f008:**
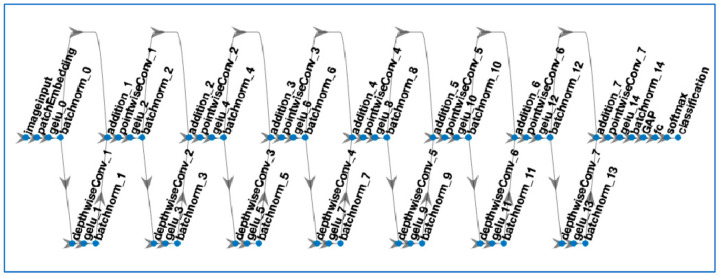
ConvMixer-based neural network used in this study.

**Figure 9 diagnostics-13-00182-f009:**
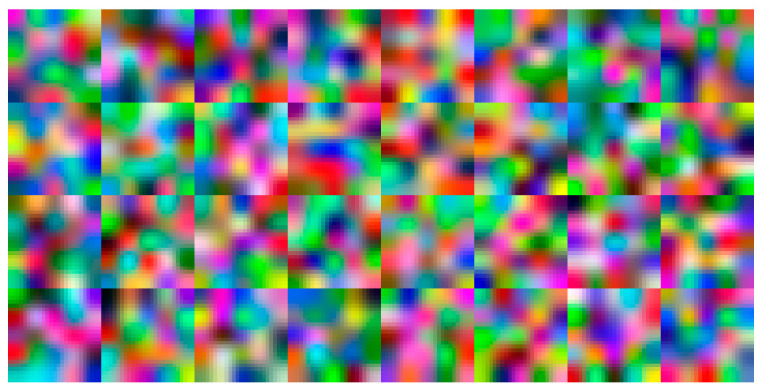
An example of ConvMixer first convolution layer weights after a training was carried out.

**Figure 10 diagnostics-13-00182-f010:**
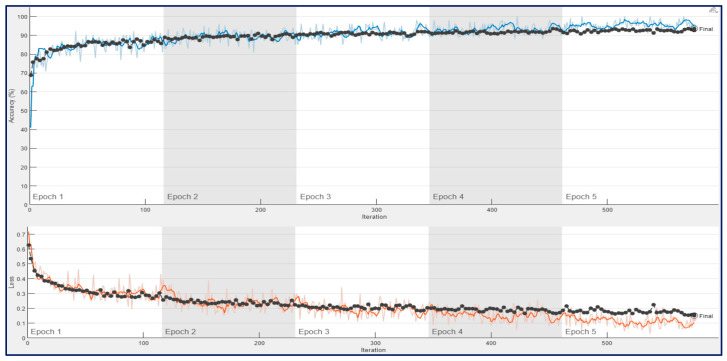
Training process result for ConvMixer with 12,800 spectrogram images in a trial on the 10-FCV.

**Figure 11 diagnostics-13-00182-f011:**
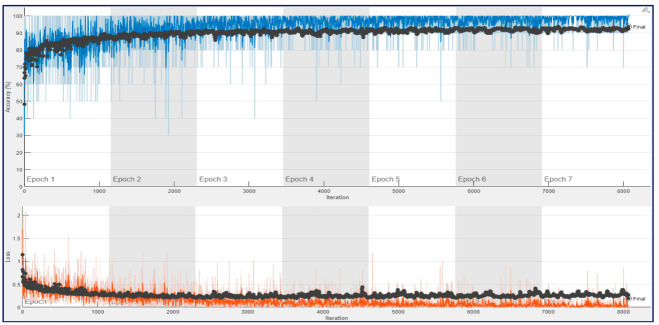
Training process result for ResNet18 with 12,800 spectrogram images in a trial on the 10-FCV.

**Figure 12 diagnostics-13-00182-f012:**
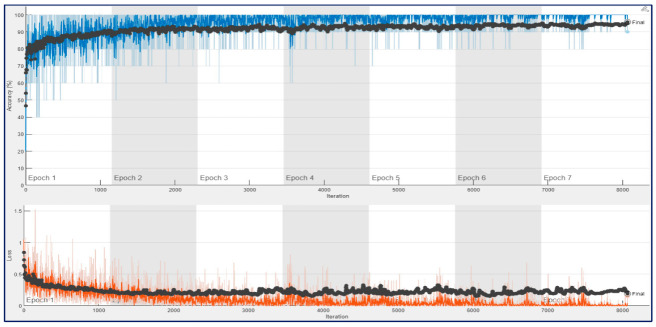
Training process result for ResNet50 with 12,800 spectrogram images in a trial on the 10-FCV.

**Figure 13 diagnostics-13-00182-f013:**
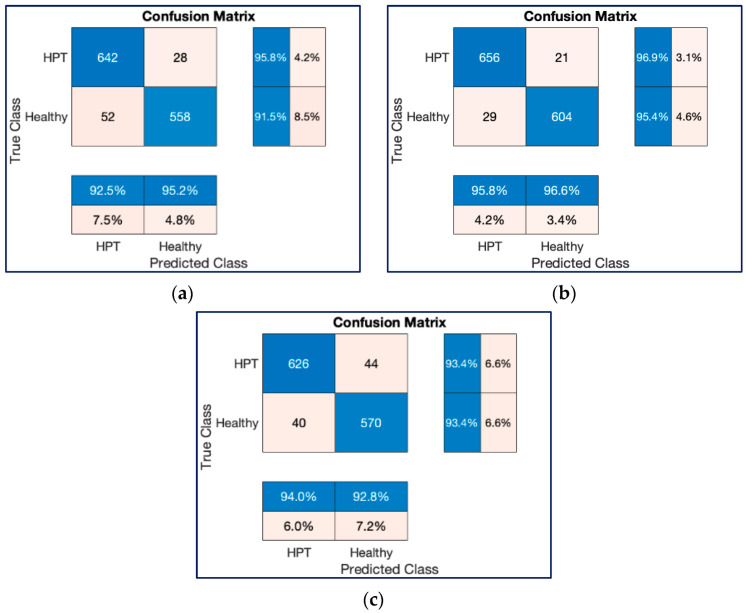
Confusion chart results for classification process with 12,800 spectrogram images in a trial on the 10-FCV. (**a**) ResNet18, (**b**) ResNet50, (**c**) ConvMixer.

**Figure 14 diagnostics-13-00182-f014:**
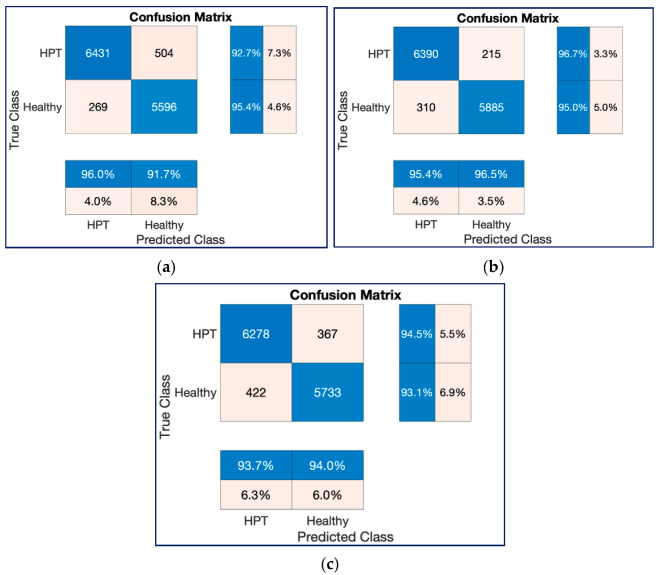
Cumulative confusion chart results for classification processes with 12,800 spectrogram images on the 10-FCV. (**a**) ResNet18, (**b**) ResNet50, (**c**) ConvMixer.

**Figure 15 diagnostics-13-00182-f015:**
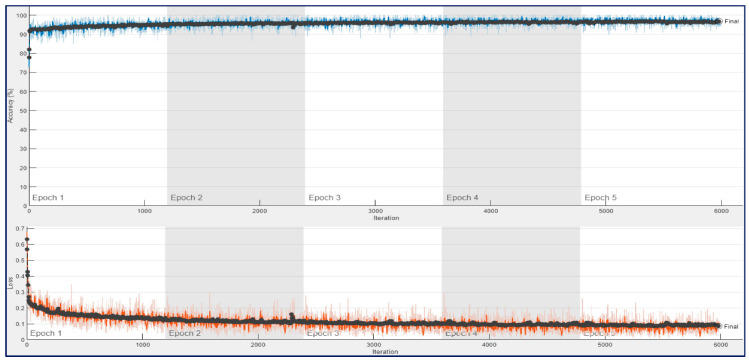
Training process result for ConvMixer with 132,968 spectrogram images in a trial on the 10-FCV.

**Figure 16 diagnostics-13-00182-f016:**
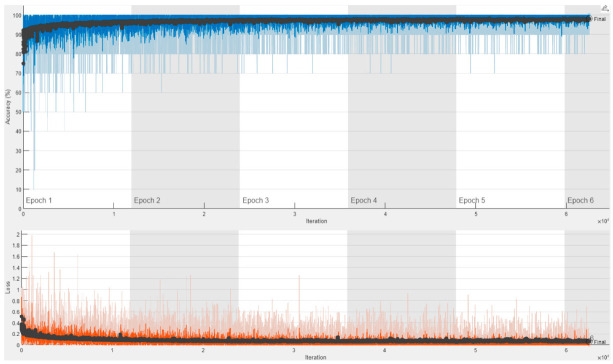
Training process result for ResNet18 with 132,968 spectrogram images in a trial on the 10-FCV.

**Figure 17 diagnostics-13-00182-f017:**
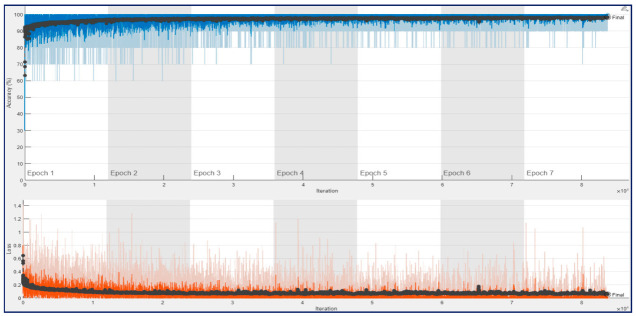
Training process result for ResNet50 with 132,968 spectrogram images in a trial on the 10-FCV.

**Figure 18 diagnostics-13-00182-f018:**
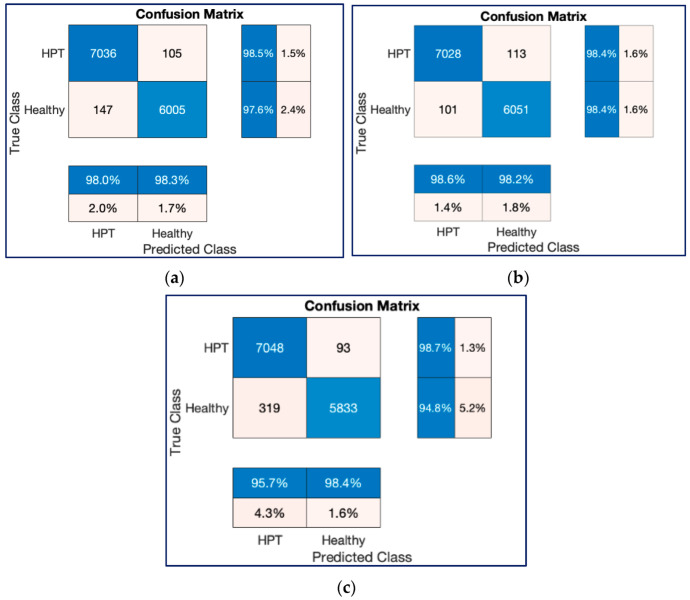
Confusion chart results for one of the classification processes with 132,968 spectrogram images in a trial on the 10-FCV. (**a**) ResNet18, (**b**) ResNet50, (**c**) ConvMixer.

**Figure 19 diagnostics-13-00182-f019:**
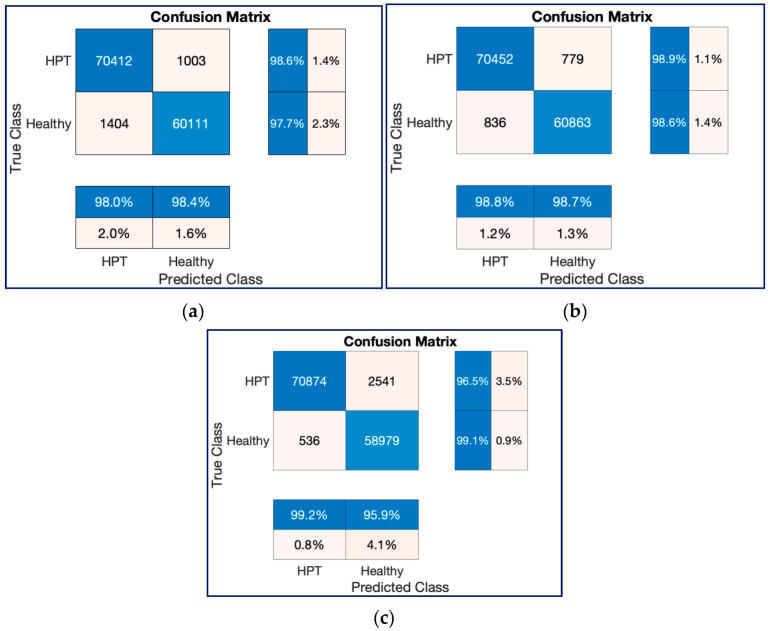
Cumulative confusion chart results for classification processes with 132,968 spectrogram images on the 10-FCV. (**a**) ResNet18, (**b**) ResNet50, (**c**) ConvMixer.

**Table 1 diagnostics-13-00182-t001:** Statistical information about the used dataset [12].

Subject Information	Hypertensive	Normotensive
Number of Subjects	61	67
Sex (Male/Female)	33/38	35/32
Age (Years)	55.6 ± 7.9	53.2 ± 9.2
Heart Rate (BPM)	77.1 ± 9.2	73.6 ± 8.3
Body Mass Index (kg/m^2^)	24.3 ± 3.6	23.7 ± 3.4
Systolic Blood Pressure (mmHg)	155.6 ± 11.2	112.1 ± 15.7
Diastolic Blood Pressure (mmHg)	103.6 ± 8.2	74.4 ± 6.3

**Table 2 diagnostics-13-00182-t002:** Average 10-FCV performance evaluation metrics for 12,800 spectrogram images.

CNN	Accuracy (%)	Precision (%)	Recall (%)	F1 Score (%)
ResNet 18	93.96	94.07	93.86	93.97
ResNet 50	95.90	95.87	95.92	95.90
ConvMixer	93.84	93.81	93.84	93.83

**Table 3 diagnostics-13-00182-t003:** Average 10-FCV performance evaluation metrics for 132,968 spectrogram images.

CNN	Accuracy (%)	Precision (%)	Recall (%)	F1 Score (%)
ResNet 18	98.18	98.20	98.16	98.18
ResNet 50	98.79	98.78	98.78	98.78
ConvMixer	97.69	97.82	97.56	97.69

**Table 4 diagnostics-13-00182-t004:** Performance comparison of the proposed work with the state-of-the-art works carried out using similar BCG dataset.

Authors	Feature/Method	Classifier	Database	Performance (%)
Song et al. [11]	Heart Rate Variability (HRV) time/Heart Beat (RR) interval, Detrended fluctuation analysis	Naïve Bayes	Their dataset (18 participants)	92.3
Liu et al. [12]	HRV time, Frequency domain feature, Sample Entropy, BCG fluctuation features	Lib Support Vector Machine, Decision tree, Naïve Bayes	Their dataset (128 participants)	84.4Acc. 82.5 Precision 85.3 Recall
Rajput et al. [13]	Cosines wavelet transform scalogram	2-D CNN	[12]	86.14
Gupta et al. [14]	Tunable Q factor wavelet transform (TQWT), Shannon entropy, log energy, Hjorth complexity, standard derivation, root mean square value, kurtosis, skewness, mean value, maximum value, and minimum value, Kruskal-Wallis	k-NN	[12]	92.21
Gupta et al. [15]	Gabor transform, smoothed pseudo-Wigner Ville distribution, short Fourier transform	Hyp-Net (CNN)	[12]	97.65
Seok et al. [16]	Hilbert transform with EMD method	CNN Regression	Their dataset (30 participants)	Standard deviation 6.24 in systolic blood pressure, 5.42 in diastolic blood pressure
Rajput et al. [17]	Empirical mode decomposition (EMD), Wavelet Transform (WT)	Ensemble gentle boost classifier, support vector machine (SVM), k-NN, decision tree	[12]	89
Proposed study	Fine-tuned spectrogram images and ConvMixer model	CNN	[12]	97.69

## Data Availability

Publicly available.

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
