# Peer review of "Automated Hypertension Detection Using ConvMixer and Spectrogram Techniques with Ballistocardiograph Signals"

_diagnostics, 2023, doi:10.3390/diagnostics13020182_

Round 1
Reviewer 1 Report
* In the abstract section the sentence “This health problem often referred to as the “silent killer,” reduces the quality of life and causes severe damage to many body parts in various ways.” The use of comma is wrong and the sentence must revise in meaning.
* In the abstract section the sentence “While creating the spectrogram images, parameters such as window type, size, overlapping rate, and fast Fourier transform size were adjusted.” "size" parameter should be written more clearly.
* In the abstract section in the sentence “The results obtained showed that the ConMixer structure gave very successful results and a very short operation time.” The word “ConMixer” should be fixed as ConvMixer.
* In the introduction section the sentence “Song et al. [11] continuously obtained BCG signals with micro motion sensitive mattress (MSM) during sleep in the dataset they created using 8 women and 10 men.” “with” should be used instead of “using”.
* Recent articles using spectrogram techniques should be cited.
For eg.
https://www.sciencedirect.com/science/article/pii/B9780323911979000126
https://ieeexplore.ieee.org/abstract/document/9393968
* In the introduction section in the sentence “They achieved a 92.3% classification accuracy using a naive Bayes classifier.” “naïve” must start with capital letter.
* In the introduction section in the sentence “They achieved a 92.21% classification accuracy rate using the K-nearest neighbor (k-NN) classifier with a 10-fold cross-validation strategy.” “K-nearest neighbor” statement should be removed because it was given above.
* In the introduction section inthe sentence “Gupta et al., in another recent study [15], developed an automatic hypertension detection system by converting BCG signals to time-frequency (T-F spectral images).” “(T-F spectral images)” should be fixed as “(T-F) spectral images.”
* The authors should add a paper outline at the end of the introduction section.
* In the Section 2-Database, at the end of sentence 2 the word “value” should be replaced with “rate”.
* In Fig. 2, picture labels must be checked.
* In section 3.4, on the first sentence the abbreviation for mlp mixer is given above, so it is unnecessary to write again.
* In section 4, after Fig. 9 the sentence “In this work, we employed two different strategies are considered. ” is unclear. It should be checked.
* The second sentence that coming after one above warning, the word dataset used as “datasets”. It should be fixed.
Reviewer 2 Report
? In the first sentence on the abstract section (line 10) there is more than one space at the beginning of the sentence.
? In the line 12, use of comma is superfluous.
? In the line 18, the parameter “size” should be corrected as “window length”.
? In the line 22 and 260, the abbreviation of the convolution mixer should be checked.
? In the line 28, the keyword hypertension should begin with lower case.
? A paper outline should be included after the introduction paragraph.
? In Figure 1, the text used in the boxes are unclear. Text puntos’ should be decrease or boxes getting bigger.
? In the line 185, the expression of “Fig.” is in bold form. Should be checked.
? In table 4 at row 6, the used classifier and its percentage performance irrelevant. The authors should check it.
? In the line 296, subtitle of Figure 9 is deficient for information. It should be noted that it shows the weights after training.
? The doi numbers in the references section should be removed.
? In the line 478, the word “propped” is meaningless. Maybe it should replace with proposed.
? In the references section, access date to the references 22 and 29 should be indicated in parentheses.
